# Synthesis of bio-based methylcyclopentadiene via direct hydrodeoxygenation of 3-methylcyclopent-2-enone derived from cellulose

Yanting Liu [1,2,6], Ran Wang [1,3,6], Haifeng Qi[1], Xiao Yan Liu [1], Guangyi Li[1], Aiqin Wang [1,4], Xiaodong Wang[1], Yu Cong [1], Tao Zhang [1,2,4✉] & Ning Li [1,4,5✉]

The exploration of highly efficient processes to convert renewable biomass to fuels and value-added chemicals is stimulated by the energy and environment problems. Herein, we describe an innovative route for the production of methylcyclopentadiene (MCPD) with cellulose, involving the transformation of cellulose into 3-methylcyclopent-2-enone (MCP) and subsequent selective hydrodeoxygenation to MCPD over a zinc-molybdenum oxide catalyst. The excellent performance of the zinc-molybdenum oxide catalyst is attributed to the formation of $ZnMoO_3$ species during the reduction of $ZnMoO_4$. Experiments reveal that preferential interaction of $ZnMoO_3$ sites with the C=O bond instead of C=C bond in vapor-phase hydrodeoxygenation of MCP leads to highly selective formations of MCPD (with a carbon yield of 70%).

[1] CAS Key Laboratory of Science and Technology on Applied Catalysis, Dalian Institute of Chemical Physics, Chinese Academy of Sciences, 116023 Dalian, China. [2] State Key Laboratory of Catalysis, Dalian Institute of Chemical Physics, Chinese Academy of Sciences, 116023 Dalian, China. [3] University of Chinese Academy of Sciences, 100049 Beijing, China. [4] iChEM (Collaborative Innovation Center of Chemistry for Energy Materials), Dalian Institute of Chemical Physics, Chinese Academy of Sciences, 116023 Dalian, China. [5] Dalian National Laboratory for Clean Energy, 116023 Dalian, China. [6] These authors contributed equally: Yanting Liu, Ran Wang. ✉email: taozhang@dicp.ac.cn; lining@dicp.ac.cn

With the increment of social concern about energy and environmental problems, the exploration of technologies for the production of fuels[1–5] and value-added chemicals[6–10] with renewable biomass has drawn a lot of attention. Methylcyclopentadiene (MCPD) is an important monomer in the production of RJ-4 fuel, a high-energy-density rocket fuel[11]. Meanwhile, it is also widely used in the synthesis of various valuable products (e.g., epoxy curing agent methylnadic anhydride (MNA), gasoline antiknock methylcyclopentadienyl manganese tricarbonyl (MMT), medicines, dye additives, organometallic catalysts, etc.)[12]. Currently, MCPD is mainly obtained from the by-products of petroleum cracking tar at a very low yield ($\sim$0.7 kg ton$^{-1}$) and high price ($\sim$10,000 USD ton$^{-1}$)[12,13]. This greatly limits its application. Previous studies have shown that the linalool can be converted to MCPD[14]. However, linalool is extracted from some special plants (such as lavender, rose, basil, and citrus aurantium, etc.) at low yields. From a practical point of view, the route for the synthesis of renewable MCPD with cheaper and more abundant biomass is highly expected. It is well-known that cellulosic biomass has the advantage of large availability, renew-ability, and $CO_2$ neutral. Therefore, the development of strategies for the production of MCPD with cellulose will be of considerable significance because of its increasing market demand (>50,000 tons year$^{-1}$) and limited petroleum resources. To the best of our knowledge, there is no report about the selective synthesis of MCPD via chemical conversion of cellulose.

Herein, we describe an approach to produce renewable MCPD from cellulose (Fig. 1). This process is an integrated technology that includes the hydrogenolysis of cellulose to 2,5-hexanedione (HD), the intramolecular aldol condensation of HD to 3-methylcyclopent-2-enone (MCP), and subsequent hydrodeoxygenation of MCP to MCPD. The first two steps have been reported by literature[15–18]. In our recent work[18], cellulose was selectively transformed into HD with a separation carbon yield of 71%. The intramolecular aldol condensation of the cellulose-derived HD produced MCP at a carbon yield of 98%. Based on these results, a high overall carbon yield of 70% MCP was obtained from cellulose. As the focus and innovation of this work, we reported the direct synthesis of MCPD by the selective hydrodeoxygenation of MCP.

## Results

**Catalytic performance of metal oxide catalysts.** As we know, the direct hydrodeoxygenation of unsaturated ketones to dienes is a reaction that has great commercial significance. However, this process is usually very challenging because the hydrogenation of C=C bond in unsaturated ketones is preferred than the hydrogenation and/or cleaving of C=O bond[19]. In the recent work of Román-Leshkov et al.[20], a series of saturated ketones and aldehydes were hydrodeoxygenated to corresponding mono-olefins (or aromatics) over various slightly reducible metal oxides (e.g., $V_2O_5$, $MoO_3$, $WO_3$, $Fe_2O_3$, and CuO). Among them, $MoO_3$ exhibited the highest hydrodeoxygenation activity. It has been suggested that the hydrodeoxygenation of ketones to olefins over $MoO_3$ follows a reverse Mars–van Krevelen mechanism, which includes the reaction of oxygen vacancy sites with the oxygenates

to yield olefinic products and the regeneration of oxygen vacancy sites by $H_2$ reduction[21]. Unfortunately, it was found that the pure $MoO_3$ is not well suited for selective hydrodeoxygenation of MCP to MCPD. As we can see from Table 1 (entry 1), $MoO_3$ catalyst suffers from a poor MCPD selectivity (18%) as a result of the excessive hydrogenation of C=C bond and the C−C bond cleavage to form various by-products (such as methylcyclopentene (MCPE), hexadienes (HDE), hexenes (HE), etc.) (Supplementary Figs. 1–6). However, it is interesting that MCPD selectivity can be improved after loading $MoO_3$ on some often used supports (e.g., ZnO, $Al_2O_3$, $ZrO_2$, and $SiO_2$) by the impregnation method (Table 1, entries 2–5). Meanwhile, such a promotion effect is more evident when the commercial nano-ZnO with a specific Brunauer–Emmett–Teller (BET) surface area of 22 m$^2$ g$^{-1}$ is used as the support. Over the $MoO_3$/ZnO catalyst, 71% MCPD selectivity was attained at a 99% MCP conversion under the optimum reaction conditions (Table 1, entry 2 and Supplementary Fig. 8). As we can see from the $H_2$ temperature-programmed reduction ($H_2$-TPR) profiles (Supplementary Fig. 7), $MoO_3$/ZnO exhibited the highest reduction temperature among the investigated catalysts. Therefore, we believe that the significantly higher MCPD selectivity over the $MoO_3$/ZnO may be attributed to the strong interaction between $MoO_3$ and ZnO support. To the best of our knowledge, this is the first report about the synthesis of MCPD via the direct hydrodeoxygenation of MCP from cellulose. Meanwhile, this work also opens up a horizon for the production of dienes with unsaturated ketone by a direct hydrodeoxygenation process.

For comparison, we also studied the catalytic performances of ZnO supported $V_2O_5$, $WO_3$, $Fe_2O_3$, and CuO catalysts under the same reaction conditions (Table 1, entries 6–9). It was noticed that the MCP conversions and MCPD carbon yields over the $V_2O_5$/ZnO, $WO_3$/ZnO, $Fe_2O_3$/ZnO, and CuO/ZnO catalysts are obviously lower than those over the $MoO_3$/ZnO catalyst. Due to this reason, we concentrated on the $MoO_3$/ZnO catalyst in the following research.

**Relationship between catalyst structure and activity.** To find out the intrinsic reason for the excellent catalytic performance of $MoO_3$/ZnO, we investigated the structure evolution of the Mo species during the preparation of the $MoO_3$/ZnO catalyst. For the $MoO_3$/ZnO catalyst precursor prepared by the impregnation of ZnO with ammonium heptamolybdate (AHM) solution, the X-ray diffraction (XRD) patterns show well-resolved peaks corresponding to the $H_3NH_4Zn_2Mo_2O_{10}$ species, as well as ZnO (Fig. 2a and Supplementary Fig. 9). Upon calcination at 600 °C, the characteristic peaks associated with $H_3NH_4Zn_2Mo_2O_{10}$ disappeared, indicating the decomposition of $H_3NH_4Zn_2Mo_2O_{10}$ to $ZnMoO_4$ (Fig. 2a and Supplementary Fig. 10). No obvious $MoO_3$ peak was detected in both the XRD patterns and Raman spectra of $MoO_3$/ZnO catalyst (Supplementary Figs. 10 and 11). The X-ray absorption near-edge structure (XANES) spectrum at the K-edge of Mo species in the catalyst exhibits the same characteristics of reference $ZnMoO_4$ compound (Fig. 2b), which is different from the AHM and $MoO_3$ compounds. These results confirm that the Mo species exists mainly as the $ZnMoO_4$ phase on the ZnO

**Fig. 1 Strategy for MCPD production from cellulose.** By the selective hydrodeoxygenation of the MCP which can be obtained from the HCl + Pd/C catalyzed cellulose hydrogenolysis followed by the MgO catalyzed intramolecular aldol condensation, a high yield of MCPD was achieved in this work.

**Table 1 Hydrodeoxygenation of MCP over various catalysts.**

| Entry | Catalyst | $T_{calcination}$ (°C)[a] | Conversion (%) | $S_{MCPD}$ (%)[b] | $S_{MCPE}$ (%)[b] | $S_{MCPO}$ (%)[b] | $S_{HDE}$ (%)[b] | $S_{HE}$ (%)[b] | $S_{Others}$ (%)[b] | Yield of MCPD (%) |
|---|---|---|---|---|---|---|---|---|---|---|
| 1 | MoO$_3$ | 600 | 98 | 18 | 37 | 2 | 17 | 10 | 16 | 18 |
| 2 | 15wt.%MoO$_3$/ZnO | 600 | 99 | 71 | 7 | 3 | 5 | 1 | 13 | 70 |
| 3 | 15wt.%MoO$_3$/Al$_2$O$_3$ | 600 | 93 | 20 | 20 | 4 | 9 | 2 | 45 | 19 |
| 4 | 15wt.%MoO$_3$/ZrO$_2$ | 600 | 91 | 32 | 2 | 6 | 2 | 1 | 57 | 29 |
| 5 | 15wt.%MoO$_3$/SiO$_2$ | 600 | 99 | 32 | 18 | 1 | 19 | 2 | 28 | 32 |
| 6 | 15wt.%V$_2$O$_5$/ZnO | 600 | 82 | 38 | 7 | 1 | 2 | 2 | 50 | 31 |
| 7 | 15wt.%WO$_3$/ZnO | 600 | 69 | 36 | 10 | 11 | 4 | 1 | 39 | 25 |
| 8 | 15wt.%Fe$_2$O$_3$/ZnO | 600 | 53 | 30 | 6 | 16 | 2 | 1 | 45 | 16 |
| 9 | 15wt.%CuO/ZnO | 600 | 77 | 27 | 9 | 16 | 2 | 1 | 45 | 21 |
| 10 | 10wt.%MoO$_3$/ZnO | 600 | 100 | 64 | 7 | 2 | 5 | 1 | 21 | 64 |
| 11 | 20wt.%MoO$_3$/ZnO | 600 | 97 | 58 | 13 | 4 | 8 | 1 | 16 | 56 |
| 12 | 15wt.%MoO$_3$/ZnO | 400 | 99 | 57 | 14 | 2 | 10 | 2 | 15 | 56 |
| 13 | 15wt.%MoO$_3$/ZnO | 500 | 99 | 68 | 8 | 3 | 6 | 2 | 13 | 67 |
| 14 | 15wt.%MoO$_3$/ZnO | 700 | 96 | 50 | 10 | 7 | 10 | 4 | 19 | 48 |
| 15[c] | 15wt.%MoO$_3$/ZnO | 600 | 0 | 0 | 0 | 0 | 0 | 0 | 0 | 0 |
| 16[d] | 15wt.%MoO$_3$/ZnO | 600 | 99 | 42 | 18 | 1 | 12 | 2 | 26 | 42 |

Reaction conditions: $T = 400$ °C, $P_{H_2} = 0.1$ MPa, weight hour space velocity (WHSV) = 0.23 (g g$^{-1}$ h$^{-1}$), the initial H$_2$/MCP molar ratio = 40, liquid products were collected after the reaction was carried out under investigated conditions for 3 h.
[a]$T_{calcination}$: the calcination temperature used in the preparation of catalyst.
[b]$S_{MCPD}$: Selectivity of methylcyclopentadiene. $S_{MCPE}$: Selectivity of methylcyclopentene. $S_{MCPO}$: Selectivity of 3-methylcyclopentanone. $S_{HDE}$: Selectivity of hexadienes. $S_{HE}$: Selectivity of hexenes. $S_{Others}$: Selectivity of the other by-products (include benzene, C$_{12}$ oligomers, and the gaseous products (e.g., CH$_4$, C$_2$H$_4$, C$_2$H$_6$ and C$_3$H$_6$ etc.)) generated during the reaction.
[c]For comparison, the reaction was carried in a nitrogen atmosphere under the same conditions as we used for the hydrodeoxygenation of MCP.
[d]The direct synthesis of MCPD using HD and H$_2$ as a feed under the same conditions as we used for the hydrodeoxygenation of MCP.

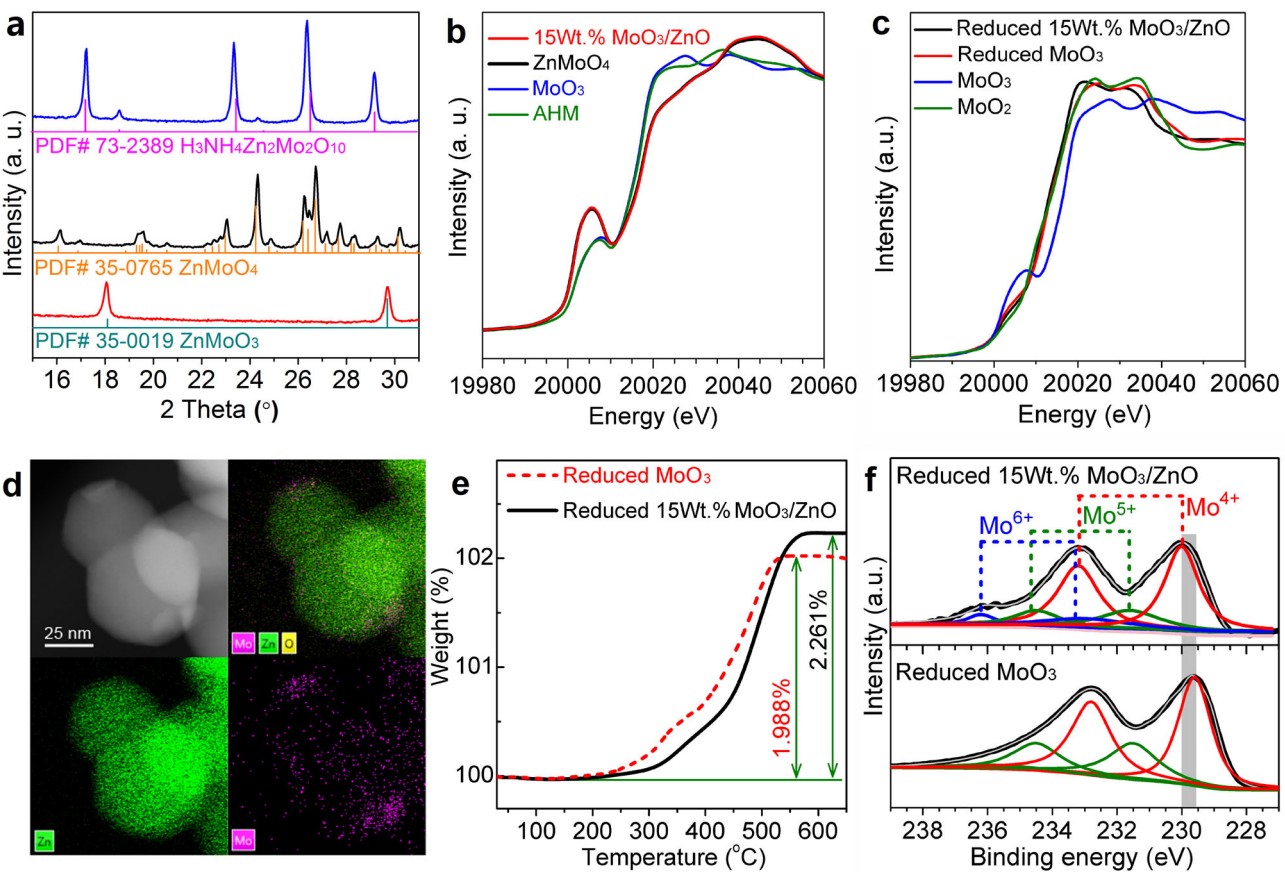

**Fig. 2 Characterization results of catalysts. a** XRD patterns of the as-prepared (blue line), calcined (black line), and reduced (red line) 15wt.%MoO₃/ZnO catalysts. **b** Normalized XANES spectra at the K-edge of Mo of the 15wt.%MoO₃/ZnO catalyst and three reference compounds (ZnMoO₄, MoO₃, and AHM). **c** Normalized XANES spectra at the K-edge of Mo of the reduced MoO₃, 15wt.%MoO₃/ZnO catalyst and two reference compounds (MoO₃ and MoO₂). **d** STEM image and elemental mappings of the reduced 15wt.%MoO₃/ZnO catalyst. **e** TG profiles of the reduced MoO₃ and 15wt.%MoO₃/ZnO catalysts in flowing air. **f** Mo 3d core level XPS spectrum of the reduced MoO₃ and 15wt.%MoO₃/ZnO catalysts. 15wt.%MoO₃/ZnO represents that the mass of Mo element in the catalyst accounts for 15% of the total mass.

support. After contacting with hydrogen under the reaction temperature of 400 °C, the XRD pattern showed that the ZnMoO₄ peaks disappeared. Meanwhile, the ZnMoO₃ peaks appeared at 18.1° and 29.7° (Fig. 2a and Supplementary Fig. 12). This result means that ZnMoO₄ was gradually transformed into ZnMoO₃ under reaction condition. The XANES spectra at the K-edge of Mo species in Fig. 2c indicated that the valance state of the Mo species in the reduced sample was close to that of the MoO₂. At the same time, the peak at the pre-edge (~20,005 eV) ascribed to the distorted octahedral structure in the MoO₃ sharply decreased, which indicated the structure of Mo species changed to the octahedral structure[22,23]. The extended X-ray absorption fine structure (EXAFS) data in *r*-space (Supplementary Fig. 13) and the data fitting results (Supplementary Table 1) showed that the average Mo-O distance in the first shell of the reduced 15wt.% MoO₃/ZnO is 2.06 Å, which is longer than those in the MoO₂ (2.02 Å) and the reduced MoO₃ (2.01 Å). Besides, the corresponding coordination number of the Mo-O of the reduced 15wt.%MoO₃/ZnO is 4.3, which is lower than that of the reduced MoO₃ (4.8). The above results indicated that the interaction between the MoOₓ and ZnO leads to the expansion of the Mo-O distance and reduction of the Mo-O coordination in the first shell. The specific BET surface area of the 15wt.%MoO₃/ZnO was measured as 19.2 m² g⁻¹, which is very close to that of ZnO support (Supplementary Table 2). Transmission electron microscopy (TEM) shows that the Mo species are uniformly dispersed on the ZnO nanoparticles (Fig. 2d).

For the deoxygenation catalyzed by the slightly reducible oxides, it is generally accepted that the oxy-compound is adsorbed on the oxygen vacancies of metal oxides[20]. Since the dissociation of C=O bonds is a high barrier process, the oxygen vacancy concentrations of the oxides will directly decide their catalytic performances in the deoxygenation reaction. The degree of generating oxygen vacancies for the oxides after reduction for 2 h were evaluated by thermogravimetry (TG) in flowing air (Fig. 2e). The percentage of weight gain for the partially reduced 15wt.%MoO₃/ZnO and MoO₃ were measured as 2.261% and 1.988%, respectively. From this result, we can see that the oxygen vacancy concentration of 15wt.%MoO₃/ZnO is higher than that of MoO₃. The interaction between the MoOₓ and ZnO might promote the formation of the oxygen vacancies, as manifested by the EXAFS results in Supplementary Fig. 13 and Supplementary Table 1: the distance and the coordination number of Mo-O in the first shell of the reduced 15wt.%MoO₃/ ZnO are longer and lower respectively than the corresponding parameters of the reduced MoO₃. If we calculated based on Mo species (In this calculation, the contribution of ZnO support was excluded because the oxygen vacancies generated by ZnO were negligible under conditions employed (Supplementary Fig. 14)), the oxygen vacancy concentrations of 15wt.%MoO₃/ZnO (10.049%) is ~5 times that of MoO₃ (1.988%). This may be the reason why 15wt.%MoO₃/ZnO catalyst is more selective for the hydrodeoxygenation of MCP to MCPD than bulk MoO₃ (Table 1, entries 1 and 2).

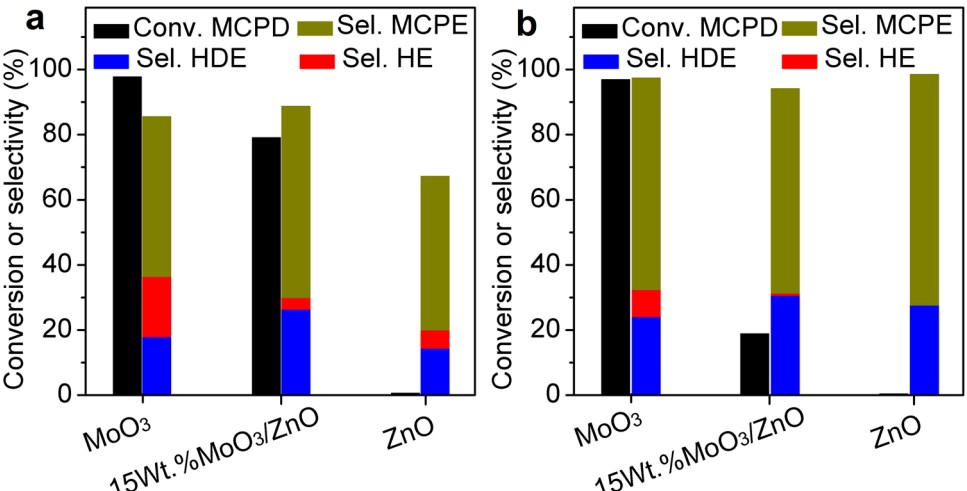

**Fig. 3 Hydrogenation performance of various catalysts. a** MCPD was used as feedstock, **b** acetone + MCPD (initial acetone/MCPD molar ratio = 1) was used as feedstock. Conditions: $T = 400\,°C$, $PH_2 = 0.1\,MPa$, WHSV = 0.23 g g$^{-1}$ h$^{-1}$, initial $H_2$/MCPD molar ratio is 40. MCPD methylcyclopentadiene, MCPE methylcyclopentene, HDE hexadienes, HE hexenes.

To further find out the reason for the high MCPD selectivity in the hydrodeoxygenation of MCP over the 15wt.%MoO₃/ZnO catalyst, the electronic properties of surface Mo species of the partially reduced catalyst were investigated by X-ray photoelectron spectroscopy (XPS) (Fig. 2f). The peak fitting suggests that there are three oxidation states (+4, +5, and +6) for Mo species on the surface of 15wt.%MoO₃/ZnO catalyst[24,25]. Mo$^{6+}$ is derived from unreduced ZnMoO₄. Small Mo$^{5+}$ peaks along with the dominant peaks of Mo$^{4+}$ can be assigned to the partially-reduced ZnMoO₄ moieties and the coordinatively unsaturated sites of ZnMoO₃, respectively. Compared with the binding energies (BE) of Mo$^{4+}$ in the reduced MoO₃ (229.5 eV), the higher BE in the reduced 15wt.%MoO₃/ZnO (230.0 eV) indicates the lower electron density of Mo species, which may hinder the hydrogenation of C=C bond and C–C bond cleavage in the hydrodeoxygenation of MCP, consequently lead to the higher MCPD selectivity (Table 1, entries 1 and 2).

**Preferential adsorption of C=O bond on catalyst**. In addition to the hydrodeoxygenation of MCP, we also examined the hydrogenation of MCPD over the 15wt.%MoO₃/ZnO, MoO₃, and ZnO catalysts to further illustrate how the electronic properties of catalysts influence their behaviors. As shown in Fig. 3a, ZnO support was almost inactive for the hydrogenation of MCPD under the investigated conditions. In contrast, both 15wt.% MoO₃/ZnO and MoO₃ are highly active for the hydrogenation of MCPD. Over them, high MCPD conversions were observed, MCPE was formed as the major product (the conversions of MCPD over the 15wt.%MoO₃/ZnO and MoO₃ are 81% and 98%, while the corresponding selectivities of MCPE are 59% and 49%, respectively). Based on this result, we cannot simply attribute the higher MCPD selectivity of MoO₃/ZnO to its lower activity for the C=C bond hydrogenation during hydrodeoxygenation of MCP. Taking into consideration that MCP has C=C bond and C=O bond simultaneously, we believe that the higher selectivity for deoxygenation of MCP to MCPD over the 15wt.%MoO₃/ZnO catalyst might be the result of energetically preferential adsorption of C=O bond in presence of C=C bond. To verify this hypothesis, acetone (a biomass-derived ketone, which has C=O bond) was co-fed with MCPD over 15wt.%MoO₃/ZnO, MoO₃, and ZnO. As shown in Fig. 3b, the co-feeding of acetone had no evident influence on the MCPD conversion over MoO₃, which means that MoO₃ catalyst remains highly active for

hydrogenation of C=C bond in MCPD even in presence of C=O bond (from acetone). In contrast, the presence of acetone significantly restrained the hydrogenation of MCPD over 15wt.% MoO₃/ZnO. This may be one reason for the higher MCPD selectivity over the 15wt.%MoO₃/ZnO catalyst for MCP hydrodeoxygenation.

In addition to acetone, the reaction with MCPD + 4-hexen-3-one (a representative of α, β-unsaturated carbonyl compound) as a reactant was investigated over the 15wt.%MoO₃/ZnO catalyst. As shown in Supplementary Fig. 15, the presence of 4-hexen-3-one significantly restrained the hydrogenation of MCPD, leading to a decrease of MCPD conversion from 79 to 17%. The phenomenon is similar to what we observed when MCPD + acetone was used as a feedstock. As we know, MCP also has C=O group. Its preferential adsorption over the 15wt.%MoO₃/ZnO catalyst may prevent the further hydrogenation of MCPD (generated from the hydrodeoxygenation of MCP) to MCPE. As the result, a high MCPD yield (or selectivity) was achieved over the MoO₃/ZnO catalyst. On the contrary, there is no such preferential adsorption over the MoO₃, which may be attributed to the lower Mo BE on partially reduced MoO₃/ZnO than that on partially reduced MoO₃ as indicated by XANES and XPS results from Fig. 2c, f, Supplementary Fig. 13 and Supplementary Table 1. As the result, the MCPD selectivity over MoO₃ is lower than that over MoO₃/ZnO.

To further simplify technology for the synthesis of MCPD with cellulose, HD (obtained by the direct hydrogenolysis of cellulose) and H₂ were also used as a feed to directly produce MCPD under the same conditions as we used for the hydrodeoxygenation of MCP. Over the optimized 15wt.%MoO₃/ZnO catalyst, 42% MCPD selectivity was attained at a 99% HD conversion (Table 1, entry 16). This result means that the intramolecular aldol condensation of HD to MCP and the subsequent selective hydrodeoxygenation of MCP to MCPD can be integrated into a one-step process, which is advantageous in a real application. To further confirm this hypothesis, we also studied the aldol condensation of HD to MCP over the 15wt.%MoO₃/ZnO catalyst using N₂ as the carrier gas. Under the same conditions as we used for the conversion of HD to MCPD under H₂ atmosphere, a 70.5% selectivity of MCP was achieved at a 53.5% HD conversion (Supplementary Table 3). To the best of our knowledge, this is the first report about the direct synthesis of a cyclic diene with straight-chain diketone as well.

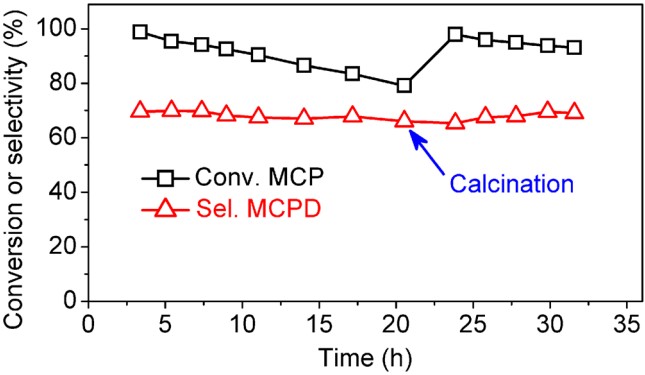

**Fig. 4 The catalytic stability of 15wt.%MoO₃/ZnO catalyst.** Conditions: $T = 400\,°C$, $PH_2 = 0.1\,MPa$, WHSV = 0.23 g g$^{-1}$ h$^{-1}$, initial H$_2$/MCP molar ratio is 40. MCP 3-methylcyclopent-2-enone, MCPD methylcyclopentadiene.

**Stability of the catalyst.** Finally, we also checked the stability of the 15wt.%MoO₃/ZnO catalyst. During the 20 h time on steam (Fig. 4), the MCPD selectivity over the 15wt.%MoO₃/ZnO catalyst kept constant, while the MCP conversion over the 15wt.%MoO₃/ZnO catalyst slightly decreased. Fortunately, such a problem can be overcome by regeneration. After being in situ calcined at 500 °C for 2 h in flowing air and then reduced at 400 °C in H₂ flow for 2 h, the catalytic activity of deactivated MoO₃/ZnO catalyst was almost restored to its initial level. Based on Supplementary Fig. 16, the spent 15wt.%MoO₃/ZnO catalyst shows similar XRD patterns as the fresh one (after in situ reduction under the reaction conditions), which may be the reason for the stable MCPD selectivity over the MoO₃/ZnO catalyst. The deactivation of MoO₃/ZnO catalyst may be attributed to the formation of coke during the reaction because of the notable weight loss during calcination at high-temperature region (380–500 °C) indicated by TG in flowing air (Supplementary Fig. 17).

In summary, we have demonstrated a facile route for the synthesis of MCPD from cellulose. Firstly, MCP was obtained at 70% overall carbon yield from cellulose via our reported method. Subsequently, as the innovation of this work, MCPD was selectively obtained by the direct hydrodeoxygenation of MCP over a 15wt.%MoO₃/ZnO catalyst. Based on the characterization results, the excellent performance of MoO₃/ZnO catalyst could be ascribed to the formation of ZnMoO₃ sites, which may preferentially adsorb C=O bond in the presence of C=C bond. As the result, a high yield of MCPD was obtained by the hydrodeoxygenation of MCP over the MoO₃/ZnO catalyst. This work enables the synthesis of renewable MCPD with cheap and abundant cellulose from a practical point of view.

## Methods

**Materials.** Analytical-grade 3-methylcyclopent-2-enone (MCP, 97%) and ammonium heptamolybdate ((NH$_4$)$_6$Mo$_7$O$_{24}$·4H$_2$O, 99%) were obtained from Shanghai Aladdin Bio-Chem Technology Co. and Tianjin Kermel Chemical Reagent Co, respectively. Commercially available nano-ZnO (ca. ~20 nm) were supplied by Nanjing Xianfeng nanometer material technology Co. LTD. Al$_2$O$_3$, ZrO$_2$, and SiO$_2$ supports were supplied by Shanghai Aladdin Bio-Chem Technology Co. The methylcyclopentadiene dimer (93%) supplied by Shanghai Aladdin Bio-Chem Technology Co. was used for calibration after distilled under 160–170 °C.

**Preparation of catalysts.** The $x$MoO$_3$/ZnO ($x$ denotes the theoretical Mo loading in weight percentage), 15wt.%MoO$_3$/Al$_2$O$_3$, 15wt.%MoO$_3$/ZrO$_2$, and 15wt.% MoO$_3$/SiO$_2$ catalysts were prepared by the impregnation method. A typical procedure was as follows: A defined amount of supports (e.g., ZnO, Al$_2$O$_3$, ZrO$_2$, and SiO$_2$) were added into an aqueous solution of (NH$_4$)$_6$Mo$_7$O$_{24}$·4H$_2$O. The mixture was stirred for 4 h, dried at 120 °C for 4 h, and calcined at 400–700 °C for 1 h in flowing air. The MoO$_3$ catalyst was prepared by the calcination of (NH$_4$)$_6$Mo$_7$O$_{24}$·4H$_2$O at 600 °C for 1 h in flowing air.

**Activity test.** The hydrodeoxygenation of 3-methylcyclopent-2-enone (MCP) was carried out under atmospheric pressure in a fixed-bed reactor. The diagram of the reaction device used in MCP hydrodeoxygenation is shown in Supplementary Fig. 18. The tubular reactor used in this work was made of 316 L stainless steel. The length, inner diameter, and constant temperature height of the reactor were measured as 30, 0.5, and 10 cm, respectively. For each test, 2.5 g of catalyst was used. The static bed layer height was measured as 2 cm. Prior to the activity test, the catalysts were activated by hydrogen (at a gas flow rate of 90 mL min$^{-1}$) at 400 °C for 2 h, and then MCP (at a liquid flow rate of 0.01 mL min$^{-1}$) was introduced into the reactor by HPLC pump along with H$_2$ (at a gas flow rate of 90 mL min$^{-1}$) which acted as a reactant and carrier gas at the same time. The contact time of MCP on the catalysts was calculated as 42.67 s.

The products passed through the reactor, cooled down to 0 °C, and became two phases in a gas-liquid separator. A small number of gas-phase products such as methylcyclopentadiene (MCPD), methylcyclopentene (MCPE), hexadienes (HDE), and hexenes (HE), were analyzed on-line by an Agilent 7890B GC after passing through the back pressure regulator. According to the concentration of feed (or specific compound) in the gas-phase effluent products (measured with the on-line GC by an external standard method), the gas flow rate of the effluent gas, and reaction time, we calculated the mole amount of feed (or specific compound) in gas-phase products. The liquid-phase products such as MCPD, MCPE, 3-methylcyclopentanone (MCPO), HDE, and hexenes HE were periodically drained from the separator and analyzed by a GC (Agilent 7890 A) fitted with a 30 m HP-5 capillary column and an FID using 1,4-dioxane as the internal standard. According to the analysis results, we calculated the mole amount of feed (or specific compound) detected in the liquid-phase products. Thus, the conversion and selectivity of the reactions were calculated based on the analysis of gas-phase and liquid-phase products. The Agilent 6540 Accurate-MS spectrometer (Q-TOF) was used for products (e.g., methylcyclopentadiene (MCPD), methylcyclopentene (MCPE), 3-methylcyclopentanone (MCPO), hexadienes (HDE) and hexenes (HE)) identification. The conversion and the selectivity of the reactions were calculated based on the following equations: Conversion (%) = 100 − total mole amount of feed detected in gas-phase and liquid-phase products/mole amount of feed pumped into reactor × 100; Selectivity for a specific compound (%) = total mole amount of a specific compound detected in the gas-phase and liquid-phase products/total mole amount of feed converted × 100.

**Characterization of catalysts.** XRD patterns were obtained on a PANalytical X'Pert-Pro diffractometer using Cu $K\alpha$ radiation ($\lambda = 1.5406$ Å) at room temperature. Data points were acquired by step scanning at a rate of 10° min$^{-1}$ from $2\theta = 10°–90°$. XRD patterns of reduced samples were tested by an in situ device. N$_2$-physisorption tests of the investigated catalysts were carried out by an ASAP 2010 apparatus. Specific surface areas of the investigated catalysts were calculated by the Brunauer–Emmett–Teller (BET) method. Average pore volumes and average pore sizes of catalysts were estimated according to the Barrett-Joyner-Halenda (BJH) method. Transmission electron microscopy (TEM) images of the samples were collected by a JEM-2100F high-resolution transmission which was operated at 200 keV. Characterizations of X-ray photoelectron spectroscopy (XPS) were conducted on an ESCALAB250xi spectrometer. Prior to the measurements, each sample was pressed into a thin disk and reduced under H$_2$ flow at 400 °C in an auxiliary pretreatment chamber. After the reduction, the obtained sample was directly introduced into the XPS chamber to avoid exposure to air. The XPS spectra were recorded at room temperature. The X-ray absorption spectra including X-ray absorption near-edge structure (XANES) and extended X-ray absorption fine structure (EXAFS) at the K-edge of Mo of the samples were collected at the BL 14W1 of Shanghai Synchrotron Radiation Facility (SSRF), China. The Mo foil was employed to calibrate the energy. The reduced samples were sealed in the glove box to protect them from contacting air. The spectra were collected at transmission mode at room temperature. The Athena software package was used to analyze the data. H$_2$-temperature-programmed reduction (TPR) tests were carried out using a Micromeritics Autochem II 2920 automated chemisorption analyzer which was connected with an online mass spectrometer (MKS Cirrus 2). 0.1 g of sample was firstly calcined in air at 400 °C for 2 h. Subsequently, the sample was cooled down in airflow to 30 °C. Finally, the sample was heated from 30 °C to 800 °C at a rate of 10 °C min$^{-1}$ in diluted hydrogen flow (10% H$_2$ in Ar, 30 mL min$^{-1}$). Thermogravimetric (TG) analysis of MoO$_3$ and 15wt.%MoO$_3$/ZnO catalysts were carried out by the TA Instrument SDT Q 600 according to the following procedure. For the fresh MoO$_3$ and 15wt.%MoO$_3$/ZnO catalyst, the sample was reduced in hydrogen at 400 °C for 2 h and swept with Ar flow at that temperature for 0.5 h. Subsequently, the sample was cooled down to 30 °C in flowing Ar. Finally, the reduced sample was heated in flowing air (100 mL min$^{-1}$) from 30 °C to 800 °C at a rate of 10 °C min$^{-1}$. For the used 15wt.%MoO$_3$/ZnO catalyst (reaction conditions: $T = 400$ °C, $PH_2 = 0.1$ MPa, WHSV = 0.23 g g$^{-1}$ h$^{-1}$, initial H$_2$/MCPD molar ratio = 40, time on stream = 20 h), the sample was directly heated in flowing air (100 mL min$^{-1}$) from 30 to 800 °C at a rate of 10 °C min$^{-1}$.

## Data availability

The data that support the findings of this study are available within the paper and its Supplementary Information and all data are available from the authors on reasonable request.

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

## Acknowledgements

This work was supported by the National Natural Science Foundation of China (nos. 21776273; 21721004; 21690082), DICP (Grant: DICP I201944), DNL Cooperation Fund, CAS (DNL180301), the Strategic Priority Research Program of the Chinese Academy of Sciences (XDB17020100), the National Key Projects for Fundamental Research and Development of China (2016YFA0202801). The authors also appreciate the help from BL 14W beamline at the Shanghai Synchrotron Radiation Facility (SSRF), Shanghai, China.

## Author contributions

Y.L. and R.W. contributed equally to this work. Y.L. and R.W. performed the catalyst preparation, characterizations, and activity tests. H.Q. collected the EXAFS data. X.L., G.L., A.W., X.W., and Y.C. analyzed the results. N.L. and T.Z. conceived the overall direction of the project. Y.L., N.L., and T.Z. co-wrote the paper. All the authors discussed the results and provided input for the manuscript.

## Competing interests

The authors declare no competing interests.
