## [Peer Review File · Nature Communications]

REVIEWER COMMENTS

Reviewer #1 (Remarks to the Author):

The hydrodeoxygenation of 3-methylcyclopent-2-enone (MCPD) to methylcyclopentadiene (MCP) is interesting as a single catalytic reaction using a new ZnMoO₃ catalyst for me. The catalyst is well characterized. However, the title of this article "Direct synthesis of MCPD via MCP from cellulose" is a conceptual proposal to industry, not science. The impact of this article depends on the market of MCP. A similar concept to this article has been published in Ref. 14. If this article is published, I don't think that the article is cited a lot. Thus, I don't think that this article is acceptable for publication in the rejection in the journal of Nature Communications.

Miscellaneous:

WHSV should be used for the space velocity instead of using LHSV.

LHSV= Hourly volumetric feed liquid flow rate/ Reaction volume.

WHSV=Hourly mass feed flow rate/ Catalyst mass.

Reviewer #2 (Remarks to the Author):

In the manuscript, the authors describe a selective reduction of biomass derived MCP into MCPD. Efficient synthesis of MCP from cellulose has been reported before by the same authors and also elsewhere. MCPD is used in larger amounts monomer in the production of rocket fuel but has also applications elsewhere. Currently, the MCPD is prepared either from petroleum cracking tar or linalool. However, both of these methods have significant limitations and a more efficient and sustainable method is desired.

The main novelty of this work is a rather selective and high yielding hydrodeoxygenation of C=O bond in the presence of a C=C bond, which remains intact. Such a conversion is hard to accomplish with existing methods.

Considering the effectiveness of the described transformation and the importance of both substrate and the product, the manuscript could be considered for publication after some revision.

The authors have thoroughly analyzed, characterized and optimized the catalyst, however, in my view there are shortcomings related mostly to the practical part which is insufficiently described.

Therefore, I recommend addressing the following:

- Fixed-bed reactor has been used. To ensure the reproducibility, please specify the size of the reactor used, the amount of catalyst used and other such relevant data.
- The MCP was introduced by HPLC pump together with H₂. How?

- It is not discussed why such flow rate (0.01 mL/min) was chosen?
- What is the contact time at such flow rate in the reactor?
- I recommend varying the flow rate/contact time and measure the effect on selectivity and conversion. Maybe a shorter contact time could boost the selectivity further and the formation of MCPE and other undesired by-products go down. As the authors have analyzed gas-phase samples on-line by GC (lines 260-262), such optimization should be rather straightforward.
- The authors test acetone+MCPD (Fig 3B) and speculate about an energetically preferential adsorption of C=O bond (line 190). What about testing MCP+MCPD (1:1) as a feed (or MCPD together with some other alpha,beta-unsaturated carbonyl compound)? What is the reactivity difference between C=O in ketone (e.g. acetone) and C=O in alpha,beta-unsaturated system (e.g. MCP)?
- Molar ratio of H₂/MCP has been 40. Could smaller excess of H₂ be used? This would be relevant in scale-up.
- Please show in SI how a value 10.049% (line 158) has been calculated.
- Grammar should be checked.

Reviewer #3 (Remarks to the Author):

Prof Li and co-workers report the synthesis of bio-based MCPD by catalytic hydrodeoxygenation of MCP, a chemical potentially derived from cellulose. Originality of this work stems from the selective C=O reduction, without concomitant C=C reduction over a 15% MoO₃/ZnO catalyst. Beside the reaction, this manuscript provides interesting insights on the 15% MoO₃/ZnO characterization, leading authors to draw hypotheses to rationalize the selectivity to MCPD. This manuscript could be published in Nat. Commun., after clarification of the following points:

- 1) The 15% MoO₃/ZnO catalyst is also capable of reducing the C=C bond of MCPD to yield MCPE, as shown by authors. Kinetically speaking, at high conversion of MCP, a competition between hydrodeoxygenation of MCP (yielding MCPD) and the hydrogenation of MCPD to MCPE should occur, as they are cascade reactions. How do you explain?
- 2) Instead of performing experiments with acetone, hydrogenation of MCPD to MCPE with incremental amount of MCP would be welcome, as it may confirm the greatest affinity of the 15% MoO₃/ZnO catalyst for C=O bond.
- 3) The title is not correct. This is not a direct route from cellulose. I would suggest "Synthesis of bio-based methylcyclopentadiene...." or "bio-based methylcyclopentadiene derived from cellulose..."
- 4) The aldolization of HD to MCP is a base-catalyzed reaction. ZnO also exhibits basic properties. Would it be possible to perform the direct reaction HD to MCPD over the 15% MoO₃/ZnO catalyst?

Response to the reviewers' comments

First of all, we appreciate all of the three reviewers for their in-depth comments and constructive suggestions. Frankly, they really helped us to improve the quality of this manuscript. We have made point-to-point responses and revised the manuscript accordingly with all changes being highlighted in the revised manuscript.

Reviewer # 1 (Remarks to the Author):

The hydrodeoxygenation of 3-methylcyclopent-2-enone (MCPD) to methylcyclopentadiene (MCP) is interesting as a single catalytic reaction using a new $ZnMoO_3$ catalyst for me. The catalyst is well characterized. However, the title of this article "Direct synthesis of MCPD via MCP from cellulose" is a conceptual proposal to industry, not science. The impact of this article depends on the market of MCP. A similar concept to this article has been published in Ref. 14. If this article is published, I don't think that the article is cited a lot. Thus, I don't think that this article is acceptable for publication in the rejection in the journal of Nature Communications.

Response: We appreciate your comments which are very helpful in improving the quality of our manuscript. For the originality of synthesis of MCPD via MCP from cellulose, we have perused the reference (Benjamin G. Harvey et al., *Solvent-free conversion of linalool to methylcyclopentadiene dimers: a route to renewable high-density fuels*, *ChemSusChem*, **2011**, 4, 465) and found that their synthetic route (see *Scheme 1*) for the production of MCPD is totally different from ours.

Scheme 1. Catalytic conversion of linalool to well-defined, renewable fuels (ChemSusChem, 2011, 4, 465).

As shown in *Scheme 1*, they used linalool as a feedstock to synthesis MCPD via a homogeneous catalytic process under the promotion of a Ru complexe. In our synthetic route, MCPD was produced by a heterogeneous catalytic process over a zinc-molybdenum oxide. There are three differences about the production of MCPD form biomass: (1) Raw materials. In their work, linalool was used as a feed. It is well known that linalool is mainly obtained from some special plants (such as lavender, rose, basil, and citrus aurantium *etc.*) based on an extraction process at low yields. In our work, MCP was used as a feedstock. MCP can be obtained from abundant cellulose at a high yield of 70% as described in “Introduction” section of the article. There are commercially practiced technologies for separation of cellulose from agriculture wastes and forest residues by the Kraft process (*Green Chem.*, **2011**, 13, 1772; *Green Chem.*, **2017**, 19, 4849); (2) Synthetic routes. In their work, MCPD was produced by the conversion of linalool to 1-methylcyclopent-2-enol and then dehydration to MCPD. In our work, MCPD was produced from MCP by a direct hydrodeoxygenation process; (3) Reaction systems. In their work, MCPD was synthesized by a homogeneous catalytic process over a Ru complexe. In our work, MCPD was synthesized by a heterogeneous catalytic process over a zinc-molybdenum oxide. Our process has an advantage of easily separating of catalyst from the reaction system over their homogeneous catalytic process. Taking into consideration of the above three reasons, our MCPD synthesis route is totally different from the previously reported one.

Following your suggestion, we have revised the initial title of “Direct Synthesis of Methylcyclopentadiene via Hydrodeoxygenation of 3-Methylcyclopent-2-enone from Cellulose” into “Synthesis of bio-based methylcyclopentadiene via direct hydrodeoxygenation of 3-methylcyclopent-2-enone derived from cellulose”. During the revising of this manuscript, we found that MCPD can also be directly obtained at a carbon yield (or selectivity) of 42% by the cascade intramolecular/hydrodeoxygenation reaction of 2,5-hexanedione (HD) (*i.e.* the feedstock used in the synthesis of MCP) over the 15wt.% MoO₃/ZnO catalyst. This result means that the intramolecular aldol condensation of HD to MCP and the selective hydrodeoxygenation of MCP to MCPD can be integrated into a one-step process, which is advantageous in real application. To the best of our knowledge, this is the first report about the direct synthesis of cyclic diene with straight-chain diketone as well. We have added the results and some comments in the Page 19 (see Table 1) and Page 9 of the revised manuscript. HD can be obtained by the direct hydrogenolysis of

cellulose. This has been reported by Essayem *et al.* (*Appl. Catal. A.*, **2015**, 504, 664) and our previous work (*Joule*, **2019**, 3, 1028). As we know, cellulose is a cheap and abundant biomass which can be easily separated from agriculture wastes and forest residues as described above. Therefore, the dependence degree of this technology on the market of MCP will be greatly decreased through the direct production of MCPD with HD. We believe that these amendments will help improve the overall competitiveness of this route.

Comment 1. Miscellaneous:

WHSV should be used for the space velocity instead of using LHSV.

LHSV= Hourly volumetric feed liquid flow rate/ Reaction volume.

WHSV=Hourly mass feed flow rate/ Catalyst mass.

Response: According to your suggestion, we have changed all of “LHSV” to “WHSV” in the Pages 19, 21, and 22 of the revised manuscript and the Pages SI-9 and SI-18 of Supporting Information.

Reviewer # 2 (Remarks to the Author):

In the manuscript, the authors describe a selective reduction of biomass derived MCP into MCPD. Efficient synthesis of MCP from cellulose has been reported before by the same authors and also elsewhere. MCPD is used in larger amounts monomer in the production of rocket fuel but has also applications elsewhere. Currently, the MCPD is prepared either from petroleum cracking tar or linalool. However, both of these methods have significant limitations and a more efficient and sustainable method is desired.

The main novelty of this work is a rather selective and high yielding hydrodeoxygenation of C=O bond in the presence of a C=C bond, which remains intact. Such a conversion is hard to accomplish with existing methods.

Considering the effectiveness of the described transformation and the importance of both substrate and the product, the manuscript could be considered for publication after some revision.

The authors have thoroughly analyzed, characterized and optimized the catalyst, however, in my view

there are shortcomings related mostly to the practical part which is insufficiently described.

Therefore, I recommend addressing the following:

Comment 1. Fixed-bed reactor has been used. To ensure the reproducibility, please specify the size of the reactor used, the amount of catalyst used and other such relevant data.

Response: Thanks for your kind reminding. The tubular reactor used in this work was made of 316L stainless steel. The length, inner diameter and constant temperature height of reactor were measured as 30 cm, 0.5 cm and 10 cm, respectively. For each test, 2.5 g of catalyst was used. The static bed layer height was measured as 2 cm. Following your constructive suggestions, we have added the above information in the Page 11 (“Activity test” section) of revised manuscript.

Comment 2. The MCP was introduced by HPLC pump together with H₂. How?

Response: In this work, MCP was introduced together with H₂ according to the method illustrated in Figure R1. To help the readers understand this, we have added this information in the Supplementary Fig. 18, Page SI-21 of Supporting Information. At the same time, we also made some modification to the “Activity test” section accordingly (see the Page 11 of the revised manuscript).

Figure R1. Diagram of the continuous flow reactor used in MCP hydrodeoxygenation.

In each test, a certain amount of catalyst was put into the tubular reactor. To keep the catalyst located at the constant temperature area, both ends of the tubular reactor were filled with quartz sand and quartz wool. After the system pressure and temperature are stabilized, the liquid reactant was pumped into the system at a certain flow rate using a high-pressure liquid chromatography pump (HPLC), accompanied with the hydrogen. The hydrogen flow rate was controlled by a mass flow controller (MFC). After coming out from the reactor and being cooled down in a trap, the product was divided into two phases through a gas-liquid separator. The gas products were analyzed online by a gas chromatography (GC) after passing through the back pressure regulator. The liquid product was taken out through the sampling valve, and then analyzed by another GC.

Comment 3. *It is not discussed why such flow rate (0.01 mL/min) was chosen?*

Response: Thank you for your reminding. According to your suggestion, we have studied the catalytic performances of 15wt.%MoO₃/ZnO under a wider range of MCP flow rate (*i. e.* WHSV) in the situation that keeping other reaction conditions unchanged, as shown below (*Table R1, Figure R2*).

Table R1. The reaction conditions for the hydrodeoxygenation of MCP.

MCP flow rate (mL min ⁻¹)	0.005	0.007	0.01	0.013	0.02
WHSV (g g ⁻¹ h ⁻¹)	0.12	0.17	0.23	0.29	0.47
Condition: 2.5 g catalyst, $T = 400\text{ }^{\circ}\text{C}$, $P_{H_2} = 0.1\text{ MPa}$, initial H ₂ /MCP molar ratio = 40.					

Figure R2. The performance of the 15wt.%MoO₃/ZnO catalyst for the hydrodeoxygenation of MCP. Reaction conditions: T = 400 °C, P_{H₂} = 0.1 MPa, initial H₂/MCP molar ratio = 40.

From *Figure R2*, we can see that the conversion of MCP decreased with the increase of WHSV. In contrast, the MCPD selectivity increased at first, reached the maximum (71%) at the WHSV of 0.23 g g⁻¹ h⁻¹, and then decreased with the further increase of WHSV. Based on these results, we chose the WHSV of 0.23 g g⁻¹ h⁻¹ (*i.e.* the flow rate of 0.01 mL min⁻¹ in this work). To help the readers understand why we chose this flow rate, these results have been added in the Supplementary Fig. 8, Page SI-9 of the updated Supporting Information section. Meanwhile, we also gave some comments in the Page 4 of revised manuscript accordingly.

Comment 4. *What is the contact time at such flow rate in the reactor?*

Response: In the reaction system, the packing volume of 2.5 g catalyst is 1.6 mL. The molar ratio of H₂/MCP used in MCP hydrodeoxygenation is 40. The reaction temperature (400 °C) is higher than the boiling point of MCP (157.5 °C). Therefore, the linear velocity of the MCP after being vaporized in tubular reactor is 41/40 time that of carrier gas (*i.e.* H₂). When H₂ flow rate is 90 mL min⁻¹, the residence time (*t*) of reactants on the catalyst is $t = \frac{\text{the packing volume of catalyst} \times 40}{(\text{the flow rate of the carrier gas of H}_2 \times 41)} = \frac{1.6 \text{ mL} \times 40}{(90 \text{ mL min}^{-1} \times 41)} = 0.017 \text{ min} = 1.041 \text{ s}$. Following your suggestion, we have added this information in the “Activity test” section of the

revised manuscript (Page 11) and the Page SI-21 of updated Supporting Information.

Comment 5. I recommend varying the flow rate/contact time and measure the effect on selectivity and conversion. Maybe a shorter contact time could boost the selectivity further and the formation of MCPE and other undesired by-products go down. As the authors have analyzed gas-phase samples on-line by GC (lines 260-262), such optimization should be rather straightforward.

Response: Following your suggestion, we studied the effect of flow rate/contact time on the MCP conversion and the product selectivities over the 15wt.%MoO₃/ZnO catalyst by just varying the WHSV and keeping other reaction conditions unchanged (see Figure R3). As you expected, the selectivity of MCPD initially increased with the increase of WHSV, reached the maximum (71%) at the WHSV of 0.23 g g⁻¹ h⁻¹, and then decreased with the further increment of WHSV (this can be comprehended because the contact time of MCP on the 15wt.%MoO₃/ZnO catalyst is insufficient at a too high WHSV. As the result, MCP cannot be converted to MCPD). The optimization of reaction conditions has been added in the Supplementary Fig. 8, Page SI-9 of the updated Supporting Information.

Figure R3. The performance of the 15wt.%MoO₃/ZnO catalyst for the hydrodeoxygenation of MCP. Reaction conditions: T = 400 °C, P_{H₂} = 0.1 MPa, initial H₂/MCP molar ratio = 40.

Comment 6. *The authors test acetone+MCPD (Fig 3B) and speculate about an energetically preferential adsorption of C=O bond (line 190). What about testing MCP+MCPD (1:1) as a feed (or MCPD together with some other α,β -unsaturated carbonyl compound)? What is the reactivity difference between C=O in ketone (e.g. acetone) and C=O in α,β -unsaturated system (e.g. MCP)?*

Response: Following your suggestion, we investigated the performances of 15wt.%MoO₃/ZnO catalyst for the hydrogenation of MCPD using the mixture of equimolar MCPD and MCP (denoted as MCPD + MCP) and the mixture of equimolar MCP and another α,β -unsaturated carbonyl compound 4-hexen-3-one (denoted as MCPD + 4-hexen-3-one) as the feedstocks.

When MCPD + MCP (the molar ratio is 1:1) was used as the feedstock, the hydrogenation of MCPD to MCPE (Reaction 1) and the hydrodeoxygenation of MCP to MCPD + MCPE (Reaction 2) took place simultaneously. These reactions are competitive. Taking into consideration of the multi roles of MCPD (both reactant and product) in the reaction system, we calculated the MCPD conversion only based on the amount of MCPD in the feedstock. According to the analysis of the product, a negative MCPD conversion (-47.4%) was obtained under the investigated condition (Due to this reason, we did not add this result into the revised manuscript). However, this result indicates that the deoxygenation of MCP to generate MCPD has priority over the hydrogenation reaction of MCPD. This is a vivid proof for the preferential adsorption of C=O bond over the 15wt.%MoO₃/ZnO catalyst in the co-presence of diene and ketene in the reaction system.

When MCPD + 4-hexen-3-one (the molar ratio is 1:1) was used as the feedstock, we can see from *Figure R4* that the presence of 4-hexen-3-one significantly restrained the hydrogenation of MCPD over 15wt.%MoO₃/ZnO, leading to the conversion of MCPD decreases from 79% to 17%. This phenomenon is similar to what we observed using MCPD + acetone as a feed. These results further confirmed that the catalyst preferentially adsorbs C=O bond in the presence of C=C bond.

We have added these results in the Supplementary Fig. 15, Page SI-18 of the updated Supporting

Information. Meanwhile, we also gave some comments in the Pages 8 and 9 of the revised manuscript accordingly.

Figure R4. Hydrogenation of MCPD, MCPD + acetone, and MCPD + 4-hexen-3-one over 15wt.% MoO₃/ZnO catalysts. Conditions: T = 400 °C, P_{H₂} = 0.1 MPa, WHSV = 0.23 g g⁻¹ h⁻¹, initial H₂/MCPD molar ratio is 40. MCPD: methylcyclopentadiene. MCPE: methylcyclopentene.

Comment 7. *Molar ratio of H₂/MCP has been 40. Could smaller excess of H₂ be used? This would be relevant in scale-up.*

Response: Thanks for your suggestion. The smaller excess of H₂ is indeed very advantageous in industrial applications. According to your suggestion, we have optimized the molar ratio of H₂/MCP for the hydrogenation of MCPD over the 15wt.%MoO₃/ZnO catalyst. As we can see from Figure R5, the 15wt.%MoO₃/ZnO catalyst exhibited the best performance at the initial H₂/MCP molar ratio of 40. We have added this result as Supplementary Fig. 8 at the Page SI-9 of updated Supporting Information. Meanwhile, we also gave some comments at the Page 4 of the revised manuscript accordingly.

Figure R5. The performance of the 15wt.%MoO₃/ZnO catalyst for the hydrodeoxygenation of MCP. Reaction conditions: T = 400 °C, P_{H₂} = 0.1 MPa, WHSV = 0.23 g g⁻¹ h⁻¹.

From *Figure R5*, we can see that both MCP conversion and MCPD selectivity over the 15wt.% MoO₃/ZnO catalyst firstly increases with increasing of the initial H₂/MCP molar ratio, the yield of MCPD reaches the maximum (70%) at the initial H₂/MCP molar ratio of 40, and then decreases with the further increment of the initial H₂/MCP molar ratio. At the lower initial H₂/MCP molar ratio, it is not conducive to the recovery of oxygen vacancies, resulting in insufficient deoxygenation capacity, lower conversion and the tendency of carbon deposits. Furthermore, the decrease of the initial H₂/MCP molar ratio will also lead to higher residence time of MCP, which would increase the probability of C=C bond hydrogenation to generate MCPO and C-C bond breakage to generate HE. After the optimization of reaction conditions, we believe that it is better to conduct the experiments at the H₂/MCP molar ratio of 40.

Comment 8. Please show in SI how a value 10.049% (line 158) has been calculated.

Response: The value of 10.049% was calculated according to the following method:

On the one hand, the mass fraction of Mo in MoO₃ is 0.67 (*i.e.* the relative atomic weight of Mo / the relative molecular weight of MoO₃ = 96 / 144 = 0.67). The oxygen vacancy concentrations of MoO₃ catalyst is 1.988%. Therefore, we can calculate the oxygen vacancy concentrations of the catalyst based on per gram Mo, *i.e.*, 1.988% / 0.67 = 2.982%. On the other hand, the Mo content of 15wt.%MoO₃/ZnO is 0.15. The oxygen vacancy concentrations of 15wt.%MoO₃/ZnO catalyst is

2.261%. Therefore, we can calculate the oxygen vacancy concentrations of the catalyst based on per gram Mo, *i.e.*, $2.261\% / 0.15 = 15.073\%$. Based on the above information, the oxygen vacancy concentrations (based on per gram Mo of catalyst) of the 15wt.%MoO₃/ZnO catalyst is 5.055 (obtained by $15.073\% / 2.982\%$) times that of MoO₃ catalyst. Finally, when the oxygen vacancy concentrations of MoO₃ catalyst is 1.988%, corresponding the oxygen vacancy concentrations of 15wt.%MoO₃/ZnO catalyst should be $1.988\% \times 5.055 = 10.049\%$.

Following your suggestion, we have added the calculation method in the Page SI-17 of Supporting Information.

Comment 9. Grammar should be checked.

Response: Thanks for your reminding. We have checked the manuscript and corrected some English grammar and typing errors.

Reviewer # 3 (Remarks to the Author):

Prof Li and co-workers report the synthesis of bio-based MCPD by catalytic hydrodeoxygenation of MCP, a chemical potentially derived from cellulose. Originality of this work stems from the selective C=O reduction, without concomitant C=C reduction over a 15% MoO₃/ZnO catalyst. Beside the reaction, this manuscript provides interesting insights on the 15% MoO₃/ZnO characterization, leading authors to draw hypotheses to rationalize the selectivity to MCPD. This manuscript could be published in Nat. Commun., after clarification of the following points:

Comment 1. *The 15%MoO₃/ZnO catalyst is also capable of reducing the C=C bond of MCPD to yield MCPE, as shown by authors. Kinetically speaking, at high conversion of MCP, a competition between hydrodeoxygenation of MCP (yielding MCPD) and the hydrogenation of MCPD to MCPE should occur, as they are cascade reactions. How do you explain?*

Response: Based on your question, we tested the activity of 15wt.%MoO₃/ZnO catalyst under different WHSV (*i.e.*, the different contact times of MCP over catalyst). To facilitate the comparison,

other reaction conditions were kept unchanged. From *Figure R6* (shown below), we can see that the MCP conversion decreased with the increase of WHSV. In contrast, MCPD selectivity first increased with the increase of WHSV, reached the maximum (71%) at the WHSV of 0.23 g g⁻¹ h⁻¹, and then decreased with the further increment of WHSV. We have added these results in the Supplementary Fig. 8, Page SI-9 of the updated Supporting Information. Meanwhile, we also gave some comments in the Page 4 of the revised manuscript accordingly.

Figure R6. The performance of the 15wt.%MoO₃/ZnO catalyst for the hydrodeoxygenation of MCP. Reaction conditions: T = 400 °C, P_{H₂} = 0.1 MPa, initial H₂/MCP molar ratio = 40.

The fixed-bed reactor simulates a plug flow state. When there are enones in the raw materials, deoxygenation reactions occur preferentially even at a high MCP conversion. However, if the reaction stream is in contact with the catalyst for too long time, the hydrogenation of MCPD will occur. Under the condition that the substrate can be completely converted, the WHSV should be increased as much as possible to reduce the contact time between the target product MCPD and the catalyst to avoid excessive hydrogenation reaction. Meanwhile, there is a balance between hydrogenation and dehydrogenation reactions at high temperatures. Under the optimal reaction conditions, MCPD with a diene structure is easily retained.

Comment 2. *Instead of performing experiments with acetone, hydrogenation of MCPD to MCPE with incremental amount of MCP would be welcome, as it may confirm the greatest affinity of the 15% MoO₃/ZnO catalyst for C=O bond.*

Response: Thanks for your reminding. Following your suggestion, we investigated the catalytic performances of 15wt.%MoO₃/ZnO under higher WHSV (*i.e.*, incremental amount of MCP). To facilitate the comparison, the other reaction conditions were kept unchanged.

Figure R7. The performance of the 15wt.%MoO₃/ZnO catalyst for the hydrodeoxygenation of MCP. Reaction conditions: T = 400 °C, P_{H₂} = 0.1 MPa, initial H₂/MCP molar ratio = 40.

As we can see from *Figure R7*, the conversion of MCP decreased with the increment of WHSV. In contrast, MCPD selectivity increased at first and then decreased. These results proved that there is a maximum affinity for C=O bond over the 15wt.%MoO₃/ZnO catalyst. As we said in the response to your second comments, the fixed-bed reactor simulates a plug flow state. When there is enone in the feedstock, deoxygenation reaction will occur preferentially even at a high MCP conversion. However, if the reaction stream is in contact with the catalyst for too long time, the hydrogenation of MCPD will occur. Under the condition that the substrate can be completely converted, the WHSV should be increased as much as possible to reduce the contact time between the target product MCPD and the catalyst to avoid excessive hydrogenation reactions. Meanwhile, there is a balance between hydrogenation and dehydrogenation reactions at high temperatures. Under the optimal reaction conditions, MCPD with a diene structure is easily retained.

To further verify the affinity of the 15wt.%MoO₃/ZnO catalyst for C=O bond, we tested its catalytic performance for the hydrogenation of the mixture of equimolar MCPD and MCP or 4-hexen-3-one (a

representative of α,β -unsaturated carbonyl compound).

When the MCPD + MCP (the molar ratio is 1:1) mixture was used as the feedstock, the hydrogenation of MCPD to MCPE and the deoxygenation of MCP (*i.e.* reactions 1 and 2) take place simultaneously. These reactions are competitive.

Taking into consideration of the multi roles of MCPD (both the reactant and the product) in the reaction system, we only calculated the MCPD conversion based on the amount of MCPD in the feedstock. It was found that the MCPD conversion we calculated was a negative value (-47.4%) under the investigated reaction conditions (Due to this reason, we did not add this result into the revised manuscript). This result indicates that the deoxygenation of MCP to MCPD takes place preferentially over the hydrogenation of MCPD. This is a vivid proof for the preferential adsorption of ketene when diene and ketene coexist in the reaction system.

Figure R8. Hydrogenation of MCPD, MCPD + acetone, and MCPD + 4-hexen-3-one over 15wt.% MoO₃/ZnO catalysts. Reaction conditions: $T = 400\text{ }^{\circ}\text{C}$, $P_{\text{H}_2} = 0.1\text{ MPa}$, $\text{WHSV} = 0.23\text{ g g}^{-1}\text{ h}^{-1}$, initial H_2/MCPD molar ratio is 40. MCPD: methylcyclopentadiene. MCPE: methylcyclopentene.

To further verify the preferential adsorption of C=O bond over the 15wt.%MoO₃/ZnO catalyst, we

used the mixture of equimolar MCPD and 4-hexen-3-one (another representative of α,β -unsaturated carbonyl compound) as the feedstock. From *Figure R8*, we can see that the presence of 4-hexen-3-one significantly restrained the hydrogenation of MCPD over 15wt.%MoO₃/ZnO. This phenomenon is similar to what we observed using the mixture of MCPD + acetone as the feedstock. This result further confirms that the catalyst preferentially adsorbs C=O bond even in the presence of C=C bond. In the revised manuscript, we have added the data in the Supplementary Fig. 15, Page SI-18 of supporting information. Meanwhile, we also gave some comments in the Page 8 of revised manuscript accordingly.

Comment 3. *The title is not correct. This is not a direct route from cellulose. I would suggest “Synthesis of bio-based methylcyclopentadiene....” or “bio-based methylcyclopentadiene derived from cellulose...”*

Response: Following your suggestion, we have changed the title of “Direct Synthesis of Methylcyclopentadiene via Hydrodeoxygenation of 3-Methylcyclopent-2-enone from Cellulose” into “Synthesis of bio-based methylcyclopentadiene via direct hydrodeoxygenation of 3-methylcyclopent-2-enone derived from cellulose”.

Comment 4. *The aldolization of HD to MCP is a base-catalyzed reaction. ZnO also exhibits basic properties. Would it be possible to perform the direct reaction HD to MCPD over the 15% MoO₃/ZnO catalyst?*

Response: This is a very intriguing and constructive suggestion. To check this possibility, we directly used HD as a raw material for the synthesis of MCPD over the optimized 15wt.%MoO₃/ZnO catalyst. Under the same conditions as we used for the hydrodeoxygenation of MCP, 42% MCPD selectivity was achieved at a 99% HD conversion, as shown in *Table R2*. This means that the intramolecular aldol condensation of HD to MCP and the selective hydrodeoxygenation of MCP to MCPD can be integrated into a one-step process. This is advantageous in real application. We have added the result in the Page 9 and the Table 1 of Page 19 of revised manuscript. In our future work, we will continue the study (including design new catalysts and optimize reaction conditions, *etc.*) to further improve the selectivity or yield of MCPD.

Table R2. Direct synthesis of MCPD from HD over the 15wt.%MoO₃/ZnO catalysts^a.

Conversion of HD (%)	Selectivity (%) ^c						Yield of MCPD (%)
	MCPD	MCPE	MCPO	HDE	HE	Others	
99.9	41.7	17.6	1.2	11.7	2.1	25.7	41.7

^a Reaction conditions: $T = 400$ °C, $P_{H_2} = 0.1$ MPa, weight hour space velocity (WHSV) = 0.23 ($\text{g g}^{-1} \text{h}^{-1}$), the initial H_2 /HD molar ratio = 40, liquid products were collected after the reaction was carried out under the investigated conditions for 3 h. ^bMCPD: methylcyclopentadiene. MCPE: methylcyclopentene. MCPO: 3-methylcyclopentanone. HDE: hexadienes. HE: hexenes. Others means the by-products generated during the reaction (include benzene, C₁₂ oligomers, and the gaseous products (*e.g.* CH₄, C₂H₄, C₂H₆, and C₃H₆ *etc.*)).

REVIEWER COMMENTS

Reviewer #1 (Remarks to the Author):

I am against publication in Nature Communications in the present form. The presented data are not mature at all. Some conflicting data are presented.

In L189-L196, the authors performed the hydrogenation of HD and successfully produced MCPD. This indicates that the catalyst has an ability for the aldol condensation of HD in N₂ flow. This should be confirmed. It is more important that the catalyst has bifunctional property of redox and base. I can't believe the W oxide species has basicity because WO₃ species is strong acid.

If the catalyst has an ability for the aldol condensation of HD, acetone could be reacted to form an aldol adduct in the experiment of Fig. 3B. In Fig. 3, acetone seems to act as a poison for the deoxygenation of MCP to MCPD. HD has a similar structure of acetone so that it may act as a poison.

In L206-L208, the deactivation of catalyst is attributed to coke formation in the decrease in weight during TG analysis. Actually, the activity was recovered after calcination in air at 500 °C. However, the carbon deposited was only 0.5 wt%. The value of 0.5 wt% is for the reaction used for either 3 h or 20 h: the reaction time is not unclear. Anyway, 0.5 wt% is so small to deactivate catalyst. I am afraid that the catalyst, ZnWO_x, seems to be sensitive to redox nature. I mean, ZnWO_x species could be further reduced during the reaction. Oxidation in air at 500 °C may recover an optimum oxidation state.

In L243, the flow rate should be clarified; gas flow rate and liquid flow rate.

In L245, the contact time was calculated using gas flow rate of hydrogen. But it should be used for reactant, MCP, not to confuse the readers.

In L247, the reaction effluent was recovered at 0 °C, and it was separated into two phases. How did the authors analyze the effluent with two phases. The mass balance, an internal standard and procedure for products analysis should also be clearly stated.

Data of 15wt%WO₃/ZnO in Fig. 3 and supplementary Fig. 15 seem to be different from those in Table 3 and supplementary Fig. 8. In supplementary Fig. 8, different selectivities seem to be presented under the same reaction conditions: the selectivities in the Fig. 8A, B, and C are different.

In supplementary Fig. 17, the reaction conditions should be addressed.

Reviewer #2 (Remarks to the Author):

Authors have addressed the comments thoroughly and adequately. The manuscript can now be published.

Reviewer #3 (Remarks to the Author):

I (reviewer 3) am satisfied by the changes made in line with my comments. In particular, I do think that the one-pot reaction brings a real added value to this work. From my side, this manuscript is now OK for publication

Response to the reviewers' comments

First of all, we appreciate again all of the three reviewers for their in-depth comments and constructive suggestions. Frankly, they really helped us to improve the quality of this manuscript. We have made point-to-point responses and revised the manuscript accordingly with all changes being highlighted in blue in the revised manuscript.

Reviewer # 1 (Remarks to the Author):

I am against publication in Nature Communications in the present form. The presented data are not mature at all. Some conflicting data are presented.

Comment 1. *In L189-L196, the authors performed the hydrogenation of HD and successfully produced MCPD. This indicates that the catalyst has an ability for the aldol condensation of HD in N₂ flow. This should be confirmed.*

Response: This is a constructive suggestion. To confirm this possibility, we investigated the aldol condensation of HD over the 15wt.%MoO₃/ZnO catalyst in N₂ flow. As you expected, 70.5% MCP selectivity was achieved at a 53.5% HD conversion under the same conditions as we used for the conversion of HD to MCPD in H₂ flow (*Table R1*). The result confirms that the 15wt.%MoO₃/ZnO can catalyze the aldol condensation of HD. We have added the result as supplementary *Table 3* in the *Page SI-17* of updated Supporting Information. Meanwhile, we also gave some comments in the *Page 9* of the revised manuscript.

Table R1. The aldol condensation of HD to MCP over the 15wt.%MoO₃/ZnO catalyst.^a

Conversion of HD (%)	Selectivity (%)			Yield of MCP (%)
	MCP	DMF	Others ^b	
53.5	70.5	8.6	20.9	37.7

^a Reaction conditions: $T = 400$ °C, $P_{N_2} = 0.1$ MPa, weight hour space velocity (WHSV) = 0.23 (g g⁻¹ h⁻¹), the initial N₂/HD molar ratio = 40. HD: 2,5-hexanedione, MCP: 3-methylcyclopent-2-enone, DMF: 2,5-dimethylfuran. ^b Others means the by-products generated during the reaction (include C₁₂

oligomers and coke).

Comment 2. *It is more important that the catalyst has bifunctional property of redox and base. I can't believe the W oxide species has basicity because WO₃ species is strong acid.*

Response: The first thing we should say is that the catalyst we used is ZnO-supported MoO₃ catalyst (15wt.% MoO₃/ZnO) instead of WO₃-based catalyst. To prove the 15wt.% MoO₃/ZnO catalyst has bifunctional property, we studied its catalytic performance for the aldol condensation of HD under N₂ flow instead of H₂ flow. As shown in *Figure R1*, 70.5% MCP selectivity was achieved over the 15wt.% MoO₃/ZnO catalyst at a 53.5% HD conversion under the same conditions as we used for direct conversion of HD to MCPD in H₂ flow, which may be attributed to the ZnO support which has higher activity for the intramolecular aldol condensation of HD (with a 100% HD conversion and 71.6% MCP selectivity under the same reaction conditions). It is well known that ZnO can be used as a solid base catalyst for some reactions (Yoshio Ono, Hideshi Hattori, *Solid Base Catalysis*. In Springer Series in Chemical Physics, Vol. 101. Toennies, Peter, Yamanouchi, Kaoru, Zinth, Wolfgang, ed. (Verlag Berlin Heidelberg and Tokyo Institute of Technology Press 2011). After taking into account the 15wt.% MoO₃/ZnO can also promote MCP hydrodeoxygenation to MCPD (this is the theme of this manuscript), we can confirm that the 15wt.% MoO₃/ZnO is indeed a bifunctional catalyst that has the ability to aldol condensation and hydrodeoxygenation.

Figure R1. The aldol condensation of HD to MCP over the ZnO and 15wt.%MoO₃/ZnO catalysts. Reaction conditions: T = 400 °C, P_{N₂} = 0.1 MPa, WHSV = 0.23 (g g⁻¹ h⁻¹), the initial N₂/reactant molar ratio = 40. HD: 2,5-hexanedione, MCP: 3-methylcyclopent-2-enone, DMF:

2,5-dimethylfuran.

Comment 3. *If the catalyst has an ability for the aldol condensation of HD, acetone could be reacted to form an aldol adduct in the experiment of Fig. 3B.*

Response: Following your suggestion, we have studied the aldol condensation of acetone over the 15wt.% MoO₃/ZnO catalyst in nitrogen atmosphere. Under the same conditions as we used for the conversion of HD to MCP in N₂ flow, only 7.7% acetone conversion was achieved over the 15wt.% MoO₃/ZnO for the aldol condensation of acetone. This value is obviously lower than the HD condensation (53.5%) over 15wt.% MoO₃/ZnO under the same reaction conditions (See *Table R2*). This result can be comprehended because the intramolecular aldol condensation of HD will lead to the five membered carbon ring which is more stable. As the result, the intramolecular aldol condensation of HD is easier than the self-aldol condensation of acetone under the investigated conditions.

Table R2. The aldol condensation of HD and acetone over the 15wt.%MoO₃/ZnO catalyst.^a

Reactant	Conversion (%)	Selectivity (%)				Yield of MCP or MO (%)
		MO	MCP	DMF	Others ^b	
HD	53.5	/	70.5	8.6	20.9	37.8
Acetone	7.7	73.7	/	/	26.3	5.7

^a Reaction conditions: $T = 400$ °C, $P_{N_2} = 0.1$ MPa, weight hour space velocity (WHSV) = 0.23 (g g⁻¹ h⁻¹), the initial N₂/reactant molar ratio = 40. MO: mesityl oxide and its isomer, HD: 2,5-hexanedione MCP: 3-methylcyclopent-2-enone, DMF: 2,5-dimethylfuran. ^b Others means the by-products generated during the reaction (include C₁₂ oligomers and coke).

Comment 4. *In Fig. 3, acetone seems to act as a poison for the deoxygenation of MCP to MCPD. HD has a similar structure of acetone so that it may act as a poison.*

Response: Thanks for your reminding. In fact, the Fig. 3 is about the hydrogenation of MCPD

instead of deoxygenation of MCP to MCPD. Theoretically, the presence of HD should decrease the conversion of MCPD, which is precisely beneficial for the direct conversion of HD to MCPD at a high yield. However, such a decrease is the results of two reasons: One reason is the poison of HD as we observed using acetone. Another reason is the cascade intramolecular aldol condensation/hydrodeoxygenation of HD will also produce MCPD (this has been proved in our manuscript). Due to the second reason, we think HD is not a good ketone to prove the poisoning effect of carbonyl group on the hydrogenation of MCPD over the 15wt.% MoO₃/ZnO catalyst. Let us suppose that HD + MCPD is used as the feedstock to prove the phenomenon, the direct conversion of HD to MCPD (*Reaction 1*) and the hydrogenation of MCPD to MCPE (*Reaction 2*) will take place simultaneously.

Taking into consideration of the multi roles of MCPD (both reactant and product) in the reaction system, we will be confused in calculating the MCPD conversion. Thus, we did not perform this experiment.

Comment 5. *In L206-L208, the deactivation of catalyst is attributed to coke formation in the decrease in weight during TG analysis. Actually, the activity was recovered after calcination in air at 500 °C. However, the carbon deposited was only 0.5 wt%. The value of 0.5 wt% is for the reaction used for either 3 h or 20 h: the reaction time is not unclear. Anyway, 0.5 wt% is so small to deactivate catalyst. I am afraid that the catalyst, ZnMoOx, seems to be sensitive to redox nature. I mean, ZnMoOx species could be further reduced during the reaction. Oxidation in air at 500 °C may recover an optimum oxidation state.*

Response: Thanks for your kind reminding. The value of 0.5 wt% is for the catalyst which has been used for 20 h. Following your suggestions, we have added the information in the Page 13 (“Characterization of catalysts” section) of the revised manuscript. Although this value is not so big as you expected, this amount of coke is enough to cover the active sites of catalyst and prevent it being contacted by reactant. As we know, the specific BET surface area of the 15wt.% MoO₃/ZnO

catalyst $19.2 \text{ m}^2 \text{ g}^{-1}$ is not so big (See supplementary Table 2 in the Page SI-16 of Supporting Information).

During catalyst regeneration, the catalyst was first oxidized in air at $500 \text{ }^\circ\text{C}$ for 2 h and then reduced at $400 \text{ }^\circ\text{C}$ in H_2 flow for 2 h (instead of directly testing the performance after oxidation in air at $500 \text{ }^\circ\text{C}$ without further pre-reduction). Sorry for our unclear description about the process for catalyst regeneration that caused your misunderstanding and come to the conclusion that oxidation in air at $500 \text{ }^\circ\text{C}$ may recover an optimum oxidation state. To help the readers understand this, we have added this information in Page 10 of the revised manuscript.

Comment 6. In L243, the flow rate should be clarified; gas flow rate and liquid flow rate.

Response: We have corrected them (see the Page 11 of the revised manuscript), as described below. Prior to the activity test, the catalysts were activated by hydrogen (at a gas flow rate of 90 mL min^{-1}) at $400 \text{ }^\circ\text{C}$ for 2 h, and then MCP (at a liquid flow rate of 0.01 mL min^{-1}) was introduced into the reactor by HPLC pump along with H_2 (at a gas flow rate of 90 mL min^{-1}) which acted as reactant and carrier gas at the same time.

Comment 7. In L245, the contact time was calculated using gas flow rate of hydrogen. But it should be used for reactant, MCP, not to confuse the readers.

Response: Following your suggestion, we have recalculated the contact time of MCP using the flow rate of MCP instead of hydrogen. In the reaction system, the packing volume of 2.5 g catalyst is 1.6 mL. The molar ratio of H_2/MCP used in MCP hydrodeoxygenation is 40. The reaction temperature ($400 \text{ }^\circ\text{C}$) is higher than the boiling point of MCP ($157.5 \text{ }^\circ\text{C}$). Therefore, when H_2 flow rate is 90 mL min^{-1} , the flow rate of the MCP after being vaporized in tubular reactor is 2.25 (*i.e.*, $90 / 40$) mL min^{-1} . The residence time (t) of MCP on the catalyst is $t = \text{the packing volume of catalyst} / \text{the flow rate of MCP after being vaporized} = 1.6 \text{ mL} / 2.25 \text{ mL min}^{-1} = 0.711 \text{ min} = 42.67 \text{ s}$. We have revised this value in the “Activity test” section of the revised manuscript (Page 11) and the Page SI-22 of updated Supporting Information.

Comment 8. In L247, the reaction effluent was recovered at $0 \text{ }^\circ\text{C}$, and it was separated into two

phases. How did the authors analyze the effluent with two phases. The mass balance, an internal standard and procedure for products analysis should also be clearly stated.

Response: In this work, the reaction product was divided into two phases through a gas-liquid separator. The gas phase products were analyzed online through a gas chromatography (GC). According to the concentration of feed (or specific compound) in the gas phase effluent products (measured with the on-line GC by external standard method), the gas flowrate of the effluent gas and reaction time, we can calculate the mole amount of feed (or specific compound) in gas phase products. The liquid product was taken out through the sampling valve, and then analyzed by another GC using 1,4-dioxane as the internal standard. According to the analysis results, we can calculate the mole amount of feed (or specific compound) detected in the liquid phase products. Thus, the conversion and selectivity of the reactions were calculated based on the analysis of gas phase products and liquid phase products according the following equations:

Conversion (%) = $100 - \frac{\text{total mole amount of feed detected in gas phase and liquid phase products}}{\text{mole amount of feed pumped into reactor}} \times 100$

Selectivity for a specific compound (%) = $\frac{\text{total mole amount of a specific compound detected in gas phase and liquid phase products}}{\text{total mole amount of feed converted}} \times 100$

Based on our analysis, the carbon balances of this work are good. In most of cases, the sum of selectivity of identified products are higher than 80%. To help the readers understand this, we have added this information in Page SI-22 of updated Supporting Information. At the same time, we also made some modification to the “Activity test” section accordingly (see the Page 12 of the revised manuscript).

Comment 9. *Data of 15wt.%MoO_x/ZnO in Fig. 3 and supplementary Fig. 15 seem to be different from those in Table 3 and supplementary Fig. 8. In supplementary Fig. 8, different selectivities seem to be presented under the same reaction conditions: the selectivities in the Fig. 8A, B, and C are different.*

Response: Thanks for your reminding. Data of 15wt.%MoO₃/ZnO in Fig. 3 and supplementary Fig.

15 are different from those in Table 3 and supplementary Fig. 8. This can be comprehended because these results were obtained with different reactants. In Fig. 3 and supplementary Fig. 15, reactants were MCPD, MCPD + acetone, and MCPD + 4-hexen-3-one, respectively. However, HD was used as the feedstock in Table 3 and supplementary Fig. 8. Therefore, it is normal for their data to be different.

Furthermore, the selectivity of MCPO indeed is different in supplementary Fig. 8. This is indeed our mistake. When we were preparing the figure, 3.0% MCPO selectivity was mistakenly type as 0.3% in Figs. 8A and 8C (See pictures below). We have corrected it. Sorry for this problem.

Previous images

Modified images

Comment 10. In supplementary Fig. 17, the reaction conditions should be addressed.

Response: Thanks for your reminding. We have added the reaction conditions ($T = 400$ °C, $P_{H_2} = 0.1$ MPa, $WHSV = 0.23$ g g⁻¹ h⁻¹, initial H₂/MCPD molar ratio = 40, time on stream = 20 h) in Fig. 17 of the Page SI-21 of updated Supporting Information.

Reviewer # 2 (Remarks to the Author):

Authors have addressed the comments thoroughly and adequately. The manuscript can now be published.

Response: Thanks again for your good comments.

Reviewer # 3 (Remarks to the Author):

I (reviewer 3) am satisfied by the changes made in line with my comments. In particular, I do think that the one-pot reaction brings a real added value to this work. From my side, this manuscript is now OK for publication.

Response: Thanks again for your good comments.

REVIEWERS' COMMENTS

Reviewer #1 (Remarks to the Author):

The authors have answered my questions, accepted most of my comments and modified the manuscript accordingly. I am satisfied with the revisions.